# Biological Credit Assignment through Dynamic Inversion of Feedforward Networks

**William F. Podlaski**[*]
Champalimaud Research
Champalimaud Centre for the Unknown
1400-038 Lisbon, Portugal

**Christian K. Machens**[*]
Champalimaud Research
Champalimaud Centre for the Unknown
1400-038 Lisbon, Portugal

## Abstract

Learning depends on changes in synaptic connections deep inside the brain. In multilayer networks, these changes are triggered by error signals fed back from the output, generally through a stepwise inversion of the feedforward processing steps. The gold standard for this process — backpropagation — works well in artificial neural networks, but is biologically implausible. Several recent proposals have emerged to address this problem, but many of these biologically-plausible schemes are based on learning an independent set of feedback connections. This complicates the assignment of errors to each synapse by making it dependent upon a second learning problem, and by fitting inversions rather than guaranteeing them. Here, we show that feedforward network transformations can be effectively inverted through dynamics. We derive this dynamic inversion from the perspective of feedback control, where the forward transformation is reused and dynamically interacts with fixed or random feedback to propagate error signals during the backward pass. Importantly, this scheme does not rely upon a second learning problem for feedback because accurate inversion is guaranteed through the network dynamics. We map these dynamics onto generic feedforward networks, and show that the resulting algorithm performs well on several supervised and unsupervised datasets. Finally, we discuss potential links between dynamic inversion and second-order optimization. Overall, our work introduces an alternative perspective on credit assignment in the brain, and proposes a special role for temporal dynamics and feedback control during learning.

## 1 Introduction

Synaptic credit assignment refers to the difficult task of relating a motor or behavioral output of the brain to the many neurons and synapses that produced it (Roelfsema and Holtmaat, 2018) — a problem which must be solved in order for effective learning to occur. While credit is assigned in artificial neural networks through the backpropagation of error gradients (Rumelhart et al., 1986; LeCun et al., 2015), a direct mapping of this algorithm to biology leads to several characteristics that are either in conflict with what is currently known about neural circuits, or that violate harder physical constraints, such as the local nature of synaptic plasticity (Grossberg, 1987; Crick, 1989).

Many biologically-plausible modifications to backpropagation have been proposed over the years (Whittington and Bogacz, 2019), with several recent studies focusing on one issue in particular, the fact that error is fed back using an exact copy of the forward weights (the "weight transport" or "weight symmetry" problem, Lillicrap et al. (2020)). Recently, it was discovered that random feedback weights are sufficient to train deep networks on modest supervised learning problems (Lillicrap et al., 2016). However, this method appears to have shortcomings in scaled-up tasks, as

---

[*]Correspondence: {william.podlaski, christian.machens}@research.fchampalimaud.org

well as in convolutional and bottleneck architectures (Bartunov et al., 2018; Moskovitz et al., 2018). Several studies have therefore aimed to identify the necessary precision of feedback (Nøkland, 2016; Xiao et al., 2018), and others have proposed to learn separate feedback connections (Kolen and Pollack, 1994; Bengio, 2014; Lee et al., 2015; Akrout et al., 2019; Lansdell et al., 2019). While it is indeed plausible that feedback weights are updated alongside forward ones, these schemes complicate credit assignment by making error backpropagation dependent upon an additional learning problem (with uncertain accuracy), and by potentially introducing more learning phases.

One important characteristic of biological neural circuits is their dynamic nature, which has been harnessed in many previous learning models (Hinton et al., 1995; O'Reilly, 1996; Rao and Ballard, 1999). Here, we take inspiration from this dynamical perspective, and propose a model of error backpropagation as a feedback control problem — during the backward pass, feedback connections are used in concert with forward connections to dynamically invert the forward transformation, thereby enabling errors to flow backward. Importantly, this inversion works with arbitrary fixed feedback weights, and avoids introducing a second learning problem for the feedback. In the following, we derive this dynamic inversion, map it onto deep feedforward networks, and demonstrate its performance on several supervised tasks, as well as an autoencoder task. Then, we discuss the biological implications of this perspective, possible links to second-order learning, and its relation to previous dynamic algorithms for credit assignment.

## 2 Deep learning in feedforward networks

### 2.1 Notation and forward transformation

We consider nonlinear feedforward networks with $L$ layers. The forward pass (forward transformation; Fig. 1a) from one layer to the next is

$$\mathbf{h}_l = g(\mathbf{a}_l) = g(\mathbf{W}_l \mathbf{h}_{l-1}), \tag{1}$$

where $\mathbf{h}_l \in \mathbb{R}^{d_l}$ is the activity of layer $l$, $g(\cdot)$ is an arbitrary element-wise nonlinearity, $\mathbf{a}_l \in \mathbb{R}^{d_l}$ is the "pre-activation" activity of layer $l$, and $\mathbf{W}_l \in \mathbb{R}^{d_l \times d_{l-1}}$ denotes the weight matrix from layers $l-1$ to $l$ (including bias). The input data, network output, and supervised target are denoted $\mathbf{x} = \mathbf{h}_0$, $\mathbf{y} = \mathbf{h}_L$, and $\mathbf{t}$, respectively. The error is denoted $\mathbf{e} = \mathbf{y} - \mathbf{t}$, and the loss function is $\mathcal{L}(\mathbf{x}, \mathbf{t})$.

### 2.2 Error backpropagation and inversion of the forward transformation

For such networks, each layer's weights are commonly optimized using gradient descent:

$$\Delta \mathbf{W}_l \propto -\frac{\partial \mathcal{L}}{\partial \mathbf{W}_l} = -\frac{\partial \mathcal{L}}{\partial \mathbf{a}_l} \frac{\partial \mathbf{a}_l}{\partial \mathbf{W}_l} = -\boldsymbol{\delta}_l \mathbf{h}_{l-1}^T, \tag{2}$$

with the backpropagated error $\boldsymbol{\delta}_l = \partial \mathcal{L} / \partial \mathbf{a}_l \in \mathbb{R}^{d_l}$. We write $\boldsymbol{\delta}_l$ in a generalized recursive form

$$\boldsymbol{\delta}_{l-1} = \frac{\partial \mathbf{a}_l}{\partial \mathbf{a}_{l-1}} \boldsymbol{\delta}_l = \mathbf{M}_l \boldsymbol{\delta}_l \circ g'(\mathbf{a}_{l-1}) = \mathbf{D}_{l-1} \mathbf{M}_l \boldsymbol{\delta}_l, \tag{3}$$

where $\circ$ is the Hadamard (element-wise) product, $\mathbf{D}_l = \mathrm{diag}(g'(\mathbf{a}_l))$, and $\mathbf{M}_l = \mathbf{W}_l^T \in \mathbb{R}^{d_{l-1} \times d_l}$ is the feedback weight matrix (the source of the weight transport problem).

As mentioned above, learning can sometimes be achieved with a fixed random feedback matrix, a strategy termed *feedback alignment* (FA), in part due to the observed alignment between the forward weights and the pseudoinverse of the feedback weights during training (Lillicrap et al., 2016). The authors of this study also describe a biologically-implausible idealization of this algorithm, *pseudobackprop* (PBP), which propagates errors through the pseudoinverse of the current feedforward weights. These results, as well as other studies proposing to learn feedback as an inverted forward transformation (e.g., target prop, Bengio (2014); Lee et al. (2015)), motivate the perspective that the goal of credit assignment is to invert the feedforward transformation of the network.

We summarize these variants of backpropagation as different choices for $\mathbf{M}_l$ in Eq. (3):

$$\mathbf{M}_l = \begin{cases} \mathbf{W}_l^T & \text{for backpropagation (BP)} \\ \mathbf{B}_l & \text{for feedback alignment (FA)} \\ \mathbf{W}_l^+ & \text{for pseudobackprop (PBP)} \end{cases} \tag{4}$$

where $\mathbf{B}_l$ is a fixed random matrix and $\mathbf{W}_l^+$ is the Moore-Penrose pseudoinverse of $\mathbf{W}_l$.

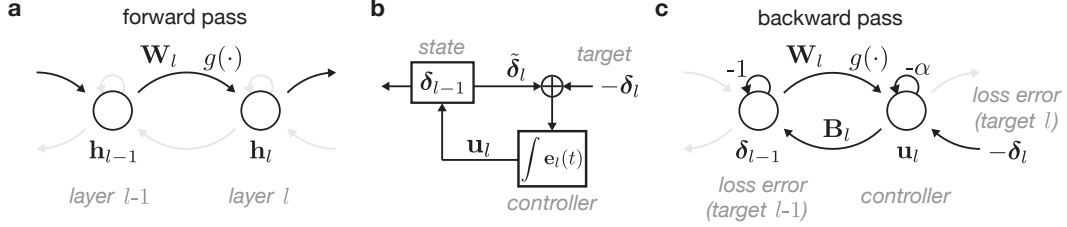

Figure 1: Schematic of forward and backward passes. **a**: Standard forward pass from Eq. (1). **b**: error propagation formulated as a feedback control problem — the difference between the forward transformation ($\tilde{\boldsymbol{\delta}}_l$) and the target error value ($\boldsymbol{\delta}_l$) is integrated and fed back to produce a new target error $\boldsymbol{\delta}_{l-1}$. **c**: Dynamic inversion during the backward pass implements this control problem.

# 3  Dynamic inversion as feedback control

We now introduce a simple recurrent architecture (Fig. 1b,c) which dynamically and implicitly performs inversions similar to those explicitly performed by pseudobackprop and target prop as outlined above. Considering the backward pass of a *linear* feedforward network (Eq. (1), with $g(x) = x$), the error from the $l$-th layer, $\boldsymbol{\delta}_l$, should be transformed into an error for the $(l-1)$-th layer, $\boldsymbol{\delta}_{l-1}$. From a linear feedback control perspective, we can let the $l$-th layer feed a control signal, $\mathbf{u}(t) \in \mathbb{R}^{d_l}$, into the $(l-1)$-th layer, such that the state of this layer, $\boldsymbol{\delta}_{l-1}$, when propagated through the feedforward transformation of the network, reproduces, as close as possible, the target error vector, $\boldsymbol{\delta}_l$, of layer $l$. We define this as a linear control problem of the following form:

$$\dot{\boldsymbol{\delta}}_{l-1}(t) = -\boldsymbol{\delta}_{l-1}(t) + \mathbf{B}_l \mathbf{u}_l(t) \tag{5}$$

$$\tilde{\boldsymbol{\delta}}_l(t) = \mathbf{W}_l \boldsymbol{\delta}_{l-1}(t), \tag{6}$$

where $\boldsymbol{\delta}_{l-1}(t) \in \mathbb{R}^{d_{l-1}}$ is the system state of layer $l-1$, $\tilde{\boldsymbol{\delta}}_l(t) \in \mathbb{R}^{d_l}$ is the readout or forward transformation of this system, $\mathbf{u}_l(t) \in \mathbb{R}^{d_l}$ is the control signal fed back from layer $l$, and $\mathbf{B}_l \in \mathbb{R}^{d_{l-1} \times d_l}$ is a matrix of arbitrary feedback weights. We define a fixed, target error value for the readout, $\boldsymbol{\delta}_l$, and a separate controller error, $\mathbf{e}_l(t) = \tilde{\boldsymbol{\delta}}_l(t) - \boldsymbol{\delta}_l$.

## 3.1  Leaky integral control

A standard approach in designing a controller is to use a proportional-integral-derivative (PID) formulation (Åström and Murray, 2010) that acts on the controller error $\mathbf{e}_l(t)$, with dynamics

$$\dot{\mathbf{u}}_l(t) = \mathbf{K}_p \dot{\mathbf{e}}_l(t) + \mathbf{K}_i \mathbf{e}_l(t) + \mathbf{K}_d \ddot{\mathbf{e}}_l(t) + \mathbf{K}_u \mathbf{u}_l(t), \tag{7}$$

where $\mathbf{K}_p$, $\mathbf{K}_i$, and $\mathbf{K}_d$ are coefficient matrices for the proportional, integral, and derivative components, respectively, along with an additional leak component with coefficients $\mathbf{K}_u$. For mathematical simplicity and biological plausibility, we only consider the integral and leak components (see Discussion for interpretation of other terms), setting their coefficients to $\mathbf{K}_i = \mathbb{I}_l$, and $\mathbf{K}_u = -\alpha\mathbb{I}_l$, where $\mathbb{I}_l$ is the identity matrix of size $d_l$. These components have a direct interpretation in rate networks (Dayan and Abbott, 2001), and have been used in other neuroscience and biological contexts (Miller and Wang, 2006; Somvanshi et al., 2015). We thus obtain the leaky integral-only controller

$$\dot{\mathbf{u}}_l(t) = -\alpha\mathbf{u}_l(t) + \mathbf{e}_l(t) = -\alpha\mathbf{u}_l(t) + \mathbf{W}_l \boldsymbol{\delta}_{l-1}(t) - \boldsymbol{\delta}_l, \tag{8}$$

which acts on Eq. (5). For a fixed target $\boldsymbol{\delta}_l$, this controller has the steady-state equality

$$\mathbf{W}_l \boldsymbol{\delta}_{l-1} = \boldsymbol{\delta}_l + \alpha\mathbf{u}_l, \tag{9}$$

which suggests that the steady state of $\boldsymbol{\delta}_{l-1}$ approximates the target $\boldsymbol{\delta}_l$ through the forward transformation (for small $\alpha$). For $\alpha > 0$, we use Eq. (5) in the steady-state to write $\boldsymbol{\delta}_{l-1}$ as

$$\boldsymbol{\delta}_{l-1} = \mathbf{B}_l(\mathbf{W}_l\mathbf{B}_l - \alpha\mathbb{I}_l)^{-1}\boldsymbol{\delta}_l = (\mathbf{B}_l\mathbf{W}_l - \alpha\mathbb{I}_{l-1})^{-1}\mathbf{B}_l\boldsymbol{\delta}_l. \tag{10}$$

When $\alpha = 0$, only one of these equalities will hold, depending on the dimensionalities $d_l$ and $d_{l-1}$. For expository purposes, we also write the solution as a function of the control signal $\mathbf{u}_l$:

$$\boldsymbol{\delta}_{l-1} = \mathbf{M}_l^{DI}(\boldsymbol{\delta}_l + \alpha\mathbf{u}_l), \tag{11}$$

where

$$\mathbf{M}_l^{DI} = \begin{cases} \mathbf{B}_l(\mathbf{W}_l\mathbf{B}_l)^{-1} & \text{for } d_l < d_{l-1} \\ \mathbf{W}_l^+ & \text{for } d_l > d_{l-1}, \ \alpha > 0 \\ \mathbf{W}_l^{-1} & \text{for } d_l = d_{l-1}, \end{cases} \tag{12}$$

and $\mathbf{W}_l^+$ is the Moore-Penrose pseudoinverse of the forward matrix $\mathbf{W}_l$. We thus see that this system dynamically inverts the forward transformation of the network (for small $\alpha$; see Suppl. Fig. 1 for plot of accuracy as a function of $\alpha$), implicitly solving the linear system $\mathbf{W}_l\boldsymbol{\delta}_{l-1} = \boldsymbol{\delta}_l$. For $d_l \geq d_{l-1}$ (expanding layer), $\mathbf{W}_l$ has a well-defined left pseudoinverse (or inverse, for $d_l = d_{l-1}$), and so the inversion follows directly from Eq. (9). In contrast, for $d_l < d_{l-1}$ (contracting layer), the system may have infinite solutions. The dynamics instead solves the fully-determined system $(\mathbf{W}_l\mathbf{B}_l - \alpha\mathbb{I})\mathbf{u}_l = \boldsymbol{\delta}_l$, which is then projected through $\mathbf{B}_l$ to obtain $\boldsymbol{\delta}_{l-1}$ (i.e., one solution to the desired linear system, constrained by $\mathbf{B}_l$).

### 3.2 Linear stability and and initialization

Dynamic inversion will only be useful if it is stable and fast. Integral-only control may exhibit substantial transient oscillations, which can be mitigated if the system dynamics are fast compared to the controller. Assuming this separation of timescales, we can study the controller dynamics from Eq. (8) when the system is at its steady state ($\boldsymbol{\delta}_{l-1} = \mathbf{B}_l\mathbf{u}_l$ from Eq. (5)):

$$\dot{\mathbf{u}}_l(t) = (\mathbf{W}_l\mathbf{B}_l - \alpha\mathbb{I}_l)\mathbf{u}_l(t) - \boldsymbol{\delta}_l. \tag{13}$$

Linear stability thus depends on the eigenvalues of $(\mathbf{W}_l\mathbf{B}_l - \alpha\mathbb{I})$. Generally, the stability of interacting neural populations (and the eigenvalues of arbitrary matrix products), is an open question and we do not aim to solve it here. We instead propose that clever initialization of $\mathbf{B}_l$ will provide stability throughout training (in addition to a non-zero leak, $\alpha$). One easy way to ensure this is to initialize $\mathbf{B}_l = -\mathbf{W}_l^T(0)$, which makes the matrix product negative semi-definite (zero index indicates the start of training). From Eq. (12), this also means that for $d_l < d_{l-1}$, dynamic inversion will use the Moore-Penrose pseudoinverse at the start of training. Note that this initialization does not imply a correspondence between the forward and backward weights throughout training, as they may become unaligned when the forward weights are updated. In the case where $d_l > d_{l-1}$, the matrix product is singular and requires $\alpha > 0$ (but see Supplementary Materials for an alternative architecture).

### 3.3 Nonlinearities

We now return to the general nonlinear case. Both nonlinear control and nonlinear inverse problems are active areas of research with solutions tailored to particular applications (Slotine et al., 1991; Mueller and Siltanen, 2012), and several approaches may be suitable here. We discuss two possibilities. First, nonlinearities may be directly incorporated into the control problem through the readout $\tilde{\boldsymbol{\delta}}(t)$ in Eq. (6), leading to a nonlinear controller with dynamics

$$\dot{\mathbf{u}}_l(t) = -\alpha\mathbf{u}_l(t) + \mathbf{W}_l g(\boldsymbol{\delta}_{l-1}(t)) - \boldsymbol{\delta}_l, \tag{14}$$

where $g(\cdot)$ is an arbitrary nonlinearity. We keep the feedback in Eq. (5) linear for simplicity. Compared to Eq. (1), the order of the matrix product and nonlinearity in (14) is reversed to obtain an error with respect to the pre-activation variables as in backpropagation. The steady-state for the controller is

$$\mathbf{W}_l g(\boldsymbol{\delta}_{l-1}) = \boldsymbol{\delta}_l + \alpha\mathbf{u}_l. \tag{15}$$

Deriving an explicit relationship between $\boldsymbol{\delta}_{l-1}$ and $\boldsymbol{\delta}_l$ is tricky here, especially with common transfer functions like tanh and ReLU, which do not have well-defined inverses (at least numerically). Again for expository purposes, we use somewhat sloppy notation and write an implicit, non-unique inverse $g^{-1}(\cdot)$, for which $g^{-1}(g(\boldsymbol{\delta}_{l-1})) \approx \boldsymbol{\delta}_{l-1}$ and $g(g^{-1}(\boldsymbol{\delta}_l)) \approx \boldsymbol{\delta}_l$. We can then write $\boldsymbol{\delta}_{l-1}$ recursively as

$$\boldsymbol{\delta}_{l-1} = g^{-1}(\mathbf{M}_l^{DI}(\boldsymbol{\delta}_l + \alpha\mathbf{u}_l)), \tag{16}$$

with $\mathbf{M}_l^{DI}$ from Eq. (12). Stability is no longer guaranteed, but in practice we find that linear stability analysis still provides a decent indication of stability in the general case.

An alternative approach for handling nonlinearities is to keep dynamic inversion linear, and then apply an element-wise multiplication by $g'(\mathbf{a}_{l-1})$ either during or after convergence (following BP, Eq. (3)). From Eqs. (10) and (11), this makes the full dynamic inversion error per layer

$$\boldsymbol{\delta}_{l-1} = \mathbf{B}_l(\mathbf{W}_l\mathbf{B}_l - \alpha\mathbb{I}_l)^{-1}\boldsymbol{\delta}_l \circ g'(\mathbf{a}_{l-1}) = (\mathbf{B}_l\mathbf{W}_l - \alpha\mathbb{I}_{l-1})^{-1}\mathbf{B}_l\boldsymbol{\delta}_l \circ g'(\mathbf{a}_{l-1}), \tag{17}$$

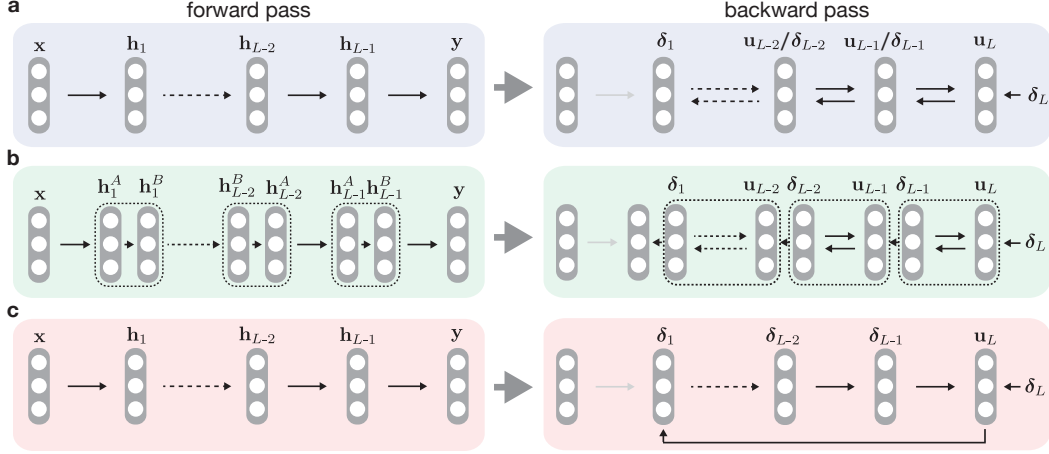

Figure 2: Schematic of forward (left) and backward (right) passes for chained dynamic inversion. **a**: *Sequential dynamic inversion* (right), in which the error is inverted through one layer at a time, with each layer first receiving control signals from the layer above, and then acting as the controller for the layer below. **B**: *Repeat layer dynamic inversion*, enabling each layer to both give and receive control, so that the full backward pass converges at once. **C**: *Single loop dynamic inversion* (SLDI) features feedback from the output layer to the first hidden layer, which effectively inverts each hidden layer.

or equivalently

$$\boldsymbol{\delta}_{l-1} = \mathbf{M}_l^{DI}(\boldsymbol{\delta}_l + \alpha \mathbf{u}_l) \circ g'(\mathbf{a}_{l-1}). \tag{18}$$

In practice we found this second option to have better performance, so we primarily use this method for the experiments presented below (except for SLDI, see next section).

## 4  Dynamic inversion of deep feedforward networks

Backpropagation in feedforward networks is a recursive, layer-wise process. However, when chaining together multiple dynamic inversions, each hidden layer must simultaneously serve as the recipient of control from the layer above, as well as the controller for the layer below. We propose three architectures which solve this problem in different ways, illustrated in Fig. 2.

### 4.1  Architectures for chained dynamic inversion

The most direct way of mapping multiple dynamic inversions onto a feedforward network is to prescribe that each inversion happens sequentially — from the output to the first hidden layer — with only one pair of layers dynamically interacting at a time (*sequential dynamic inversion*, Fig. 2a). Such a scheme begins by feeding the output error, $\boldsymbol{\delta}_L$, into the output layer, which provides control to the last hidden layer until convergence to the target $\boldsymbol{\delta}_{L-1}$. Next, this target is held fixed and is re-passed as input back into layer $L-1$, which now acts as a controller for layer $L-2$, to obtain the target $\boldsymbol{\delta}_{L-2}$. This is repeated until the first hidden layer converges to its target, $\boldsymbol{\delta}_1$. This scheme requires a backward pass with multiple steps for networks with more than one hidden layer ($L-1$ steps).

The fact that each hidden layer functions as both a recipient of control, and a controller itself, motivates the second architecture, in which the hidden layers have two separate populations, each serving one of these roles (*repeat layer dynamic inversion*, Fig. 2b). For the forward pass to remain unchanged, these populations ($\mathbf{h}_l^A$ and $\mathbf{h}_l^B$) have an identity transformation between them. During the backward pass, each controller receives the target value $\boldsymbol{\delta}_l$ as it settles, speeding up convergence. The steady state errors will be equivalent to the sequential case, but only a single backward pass is needed. Due to the equivalence of this scheme to the first, we do not explicitly simulate it here.

An alternative approach to chaining multiple dynamic inversion control problems together is to turn them into a single problem (*single loop dynamic inversion*, SLDI, Fig. 2c). In this scheme, the output layer acts as the controller for the activity of the first hidden layer, and the forward transformation

encompasses all layers in between (see Supplementary Materials for a detailed description). While in some special cases this scheme is equivalent to the ones above, in general the dynamics will converge to a different solution. We test the single loop backward pass on one experiment below (nonlinear regression), but mainly focus on the sequential method.

## 4.2 Update rules

We define the backpropagated error signal $\boldsymbol{\delta}_l$ for dynamic inversion (DI) as the steady state of the linearized feedback control dynamics from Eq. (17), and weight updates as in Eq. (2). Biases are not included in the dynamics of the backward pass, but are updated with the layer-wise error signals similar to standard backprop. As a point of comparison, we also implement a non-dynamic inversion (NDI), in which the exact steady state from Eq. (17) is used in lieu of simulating temporal dynamics. Therefore, correspondence between DI and NDI updates is indicative of successful convergence of the dynamics. In contrast, single-loop dynamic inversion (SLDI) utilizes a nonlinear controller as in Eq. (14), then weight updates as in Eq. (2) (Supplementary Materials).

## 4.3 Relation to second-order learning

The inversion of the forward weights in DI suggests a resemblance to second-order learning (Martens, 2014; Lillicrap et al., 2016). Though a full theoretical study is out of the scope of this paper, in the Supplementary Materials, we postulate a link to layer-wise Gauss-Newton optimization (Botev et al., 2017), and describe a simple example. Interestingly, recent work linking target propagation to Gauss-Newton optimization shows that the dynamic inversion of *targets* rather than errors may produce a more coherent connection to second-order learning (Meulemans et al., 2020; Bengio, 2020). Specifically, Meulemans et al. (2020) propose that a direct feedback approach similar to a single-loop architecture applied to each hidden layer (Fig. 2c) may be most effective (see Discussion).

# 5 Experiments

We tested dynamic inversion (DI) and non-dynamic inversion (NDI) against backpropagation (BP), feedback alignment (FA), and pseudobackprop (PBP) on four modest supervised and unsupervised learning tasks — linear regression, nonlinear regression, MNIST classification, and MNIST autoencoding. We varied the leak values ($\alpha$) for DI and NDI, as well as the feedback initializations ("Tr-Init", $\mathbf{B}_l = -\mathbf{W}_l^T$; "R-Init", random stable $\mathbf{B}_l$) for DI, NDI, and FA. To impose stability for random initialization, we optimized the feedback matrix $\mathbf{B}_l$ using smoothed spectral abscissa (SSA) optimization (Vanbiervliet et al., 2009; Hennequin et al., 2014) at the start of training (Supplementary Materials). We note that DI remained stable throughout training for all experiments, suggesting that initialization is sufficient to ensure stability. DI was simulated numerically using 1000 Euler steps with $dt = 0.5$.

## 5.1 Linear and nonlinear function approximation

Following Lillicrap et al. (2016), we tested the algorithms on a simple linear regression task with a two-layer network (dim. 30-20-10). Training was done with a fixed learning rate (Fig. 3a), or with fixed norm weight updates (Fig. 3b) in order to probe the update directions that each algorithm finds. All algorithms were able to solve this simple task with ease, with DI, NDI, and PBP converging faster than BP and FA in the fixed norm case, suggesting they find better update directions. Dynamic inversion remained stable throughout training for all examples shown, with updates well-aligned to the non-dynamic version (Fig. 3c,d). Furthermore, the alignment between the feedback and the negative transpose of the forward weights settled to around 45 degrees for all DI and NDI models, which also produced alignment with the PBP updates (Fig. 3e,f).

Next, we tested performance on a small nonlinear regression task with a three-layer network (dim. 30-20-10-10, tanh nonlinearities) also following Lillicrap et al. (2016). All inversion algorithms finished with equal or better performance compared to BP and FA (Fig. 3g), but often with slower convergence, which was unexpected considering the potential link to second-order optimization (see Discussion). DI dynamics were stable throughout training, remaining nearly constant (Fig. 3h). Furthermore, DI again closely followed NDI updates (Fig. 3i), though error curves between the two drifted apart during learning (Fig. 3g), indicating that a small amount of variability in convergence can

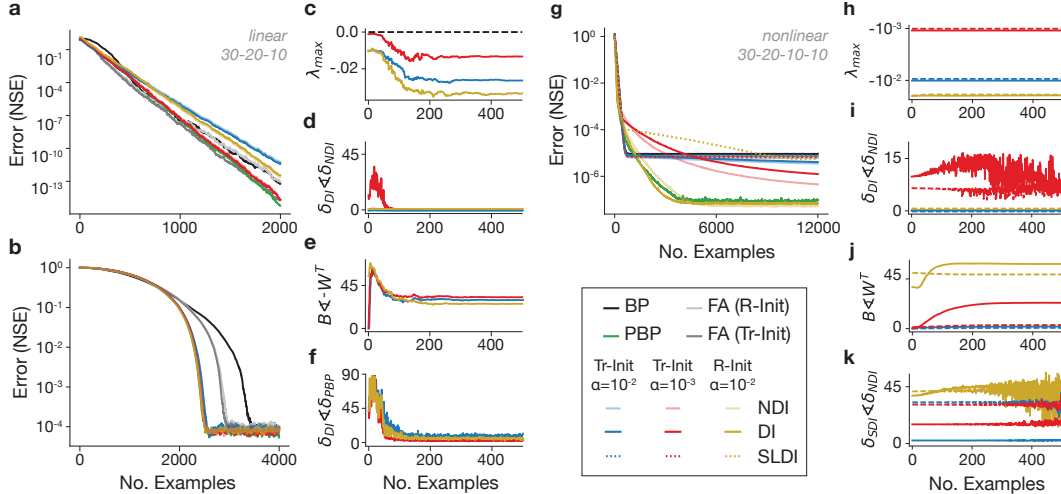

Figure 3: Results for linear (30-20-10) and nonlinear (30-20-10-10) regression tasks for BP, FA, PBP, and three realizations of DI, NDI, and SLDI (only for nonlinear) with different weight initialization and leak values. Legend below panel **g**. Learning rate $=10^{-2}$ for all algorithms. **a**: Normalized mean squared error (NMSE) for linear regression. **b**: NMSE for linear regression with fixed-norm weight updates. **c**: Stability of DI as measured by the maximum real eigenvalues of the system dynamics. **d**: Angle between the backpropagated error vectors to the hidden layer ($\delta_1$) for NDI vs DI. **e**: Angle between feedback weights and negative transpose of forward weights (shown for DI and NDI). **f**: Angle between backpropagated error vectors for DI and PBP. **g-j**: Same as **a,c,d**, **e** but for nonlinear regression. **f**: Angle between the backpropagated error vectors to the hidden layer for NDI vs SLDI. Dashed lines refer to output layer, and solid linear refer to middle layer in **h-k**.

lead to different learning trajectories. Alignment of feedback **B** varied with the layer and algorithm (Fig. 3j) — some layers settled to ~45 degrees, but others remained close to zero — this is intriguing, but may be due to the simplicity of the problem. Finally, we also simulated single-loop dynamic inversion (SLDI). The alignment between SLDI and NDI was small (~45 degrees or less), but larger than DI, supporting the claim that it converges to similar, but not necessarily equivalent steady states (Fig. 3k).

## 5.2 MNIST classification and autoencoder

We next tested dynamic inversion on the MNIST handwritten digit dataset, where we use the standard training and test datasets (LeCun et al., 1998), with a two-layer architecture (dim. 784-1000-10 as in Lillicrap et al. (2016)). All algorithms showed decent performance after 10 epochs (Fig. 4a), but with DI flattening out at a higher test error (BP, 2.2%; FA, 1.8%; PBP, all NDI, DI ~2.8%), again suggesting slower convergence (Discussion). We speculate that a more thorough exploration of hyperparameters, such as changing the architecture or using mini-batches may help here.

Finally, we trained a bottleneck autoencoder network on the MNIST dataset (dim. 784-500-250-30-250-500-784; nonlinearities tanh-tanh-linear-tanh-tanh-linear, a reduced version of Hinton and Salakhutdinov (2006)) with mini-batch training (100 examples per batch). Notably, first-order optimization algorithms have trouble dealing with the "pathological curvature" of such problems and often have very slow learning (Martens, 2010) (especially FA, Lansdell et al. (2019)). We trained the algorithms with random uniform weight initializations, similar to the previous experiments, as well as with random orthogonal initializations, which has been shown to speed up learning (Saxe et al., 2013). We found that BP only learns successfully with orthogonal weight initialization, whereas PBP, NDI, and DI perform decently in either case, further suggesting they use second-order information. Notably, PBP, NDI, and DI performance is slower with orthogonal initialization, where second-order information is not useful (but this might be mitigated by having non-orthogonal feedback weights). Furthermore, FA performed poorly in both cases, and regardless of the type of feedback initialization. This provides evidence that dynamic inversion can be superior to random feedback in some tasks, and

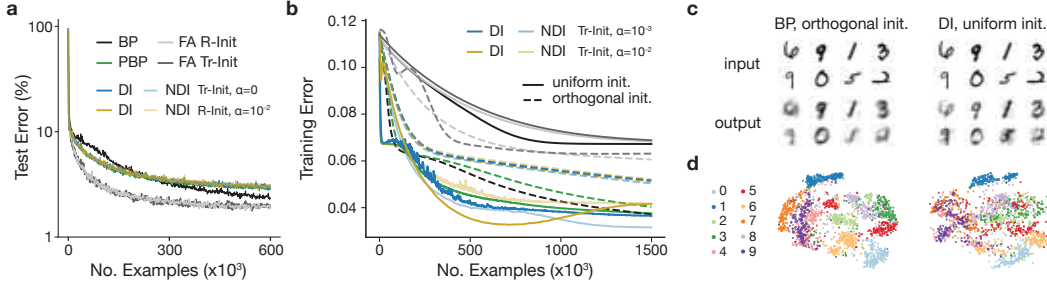

Figure 4: Results for MNIST classification (784-1000-10 architecture) and autoencoding (784-500-250-30 architecture) trained with BP, FA, PBP, and two realizations of NDI and DI. **a**: Test error of MNIST classification over 10 epochs of online training (learning rate=$10^{-3}$ for all algorithms). **b**: Training error of MNIST autoencoding over 25 epochs of mini-batch training (learning rate = $10^{-6}$ for all algorithms). **c,d**: Examples of input and output digits and a 2D t-SNE representation of the 30-dim. latent space for BP (orthogonal init.) and DI (uniform init.).

that weight initialization alone is not responsible for this benefit. BP with orthogonal initialization and DI with random initialization result in similar performance (Fig. 4c,d).

# 6   Discussion

## 6.1   Related work

Several previous studies have proposed biologically-plausible solutions to credit assignment involving dynamics of some kind, but most are conceptually distinct from our framework — e.g., learning using differences in activity over time or phase (contrastive Hebbian learning, O'Reilly (1996); Scellier and Bengio (2017)), using explicit error-encoding neurons (Whittington and Bogacz, 2017), or incorporating dendritic compartments (Guerguiev et al., 2017; Sacramento et al., 2018; Payeur et al., 2020). Furthermore, the aim of most of these models is to approximate error *gradients*, whereas dynamic inversion may relate more to second-order optimization. Most relevant to our work is a recent paper which also proposes to re-use forward weights in order to propagate errors through a feedback loop (Kohan et al., 2018). However, the authors do not formulate this as an inversion of the forward pass, and use a contrastive learning scheme that differs substantially from what we do here.

Dynamic inversion offers a novel solution to the weight transport problem, of which other solutions exist, such as using learned feedback weights (Kolen and Pollack, 1994; Akrout et al., 2019; Lansdell et al., 2019). However, in contrast to previous approaches, we frame credit assignment as a control problem. From this perspective, dynamic inversion can be seen as feedback control, whereas backpropagation and learned feedback are examples of feed-forward or predictive control (Jordan, 1996). We do not claim that dynamic inversion is superior or more plausible than learning feedback (though each may have advantages), and a thorough comparison of these different solutions would be merited. Furthermore, dynamic inversion does not address other implausibilities of backpropagation, such as the use of separate learning phases and signed errors (Lillicrap et al., 2020). In principle, dynamic inversion may be combined with insights from other models to address these other problems. For example, target propagation also aims to learn (non-dynamic) inversions (Bengio, 2014; Lee et al., 2015), making it conceptually similar — in principle dynamic inversion can also be used to propagate targets, which may afford additional biological plausibility or, as mentioned above, a more direct relationship to second-order optimization (Meulemans et al., 2020; Bengio, 2020).

## 6.2   Biological implications

Unlike many other biologically-plausible algorithms for credit assignment, dynamic inversion does not require precise feedback weights. This may be an important distinction, as it not only relaxes the assumptions on feedback wiring, but could also allow for feedback to be used concurrently for other roles, such as attention and prediction (Gilbert and Li, 2013), though we do not verify this here. The proposed architectures for chained dynamic inversion (Fig. 2) suggest different ways of using feedback for learning, and even leaves the possibility for direct feedback to much lower areas which

is known to exist in the brain (Felleman and Van Essen, 1991). Additionally, our work depends upon the stability and control of recurrent dynamics between interacting populations (or brain areas), which has received recent interest in neuroscience (Joglekar et al., 2018). Stability and fast convergence of dynamic inversion requires slow control (Eq. (13)), implying that higher-order areas should be slower than the lower areas they control. Indeed, activity is known to slow down as it moves up the processing hierarchy (Murray et al., 2014), which fits with this picture.

### 6.3 Limitations and future work

We see two main limitations of dynamic inversion as a model for credit assignment in the brain. First, DI requires stable dynamics and time to converge. Both the maintenance of stability during learning (Keck et al., 2017) and "initialization" of weights during brain development (Zador, 2019) are hypothesized to be important in biological networks, but many open questions remain. Furthermore, fast convergence would be easier to achieve with full PID control, as in Eq. (7) (Åström and Murray, 2010). Spike-based representations could help here since they effectively add a derivative component to the signal (Eliasmith and Anderson, 2004; Boerlin et al., 2013; Abbott et al., 2016).

Second, due to the relationship of our scheme with second-order learning, dynamic inversion could share some of the same problems (Martens, 2014; Kunstner et al., 2019). This may include overly small, "conservative" updates (Martens, 2014) or attraction to saddle points (Dauphin et al., 2014), thus helping to explain the slow convergence observed here. Furthermore, as shown in Meulemans et al. (2020), utilizing approximate inversions may lead to update directions that are primarily in the null space of the network output, which could be a problem in using DI for contracting layers. While our method avoids the cost of explicitly calculating and inverting a Hessian or Gauss-Newton matrix, in common with standard second-order methods (Pearlmutter, 1994; Schraudolph, 2002; Martens, 2010), its performance will be enhanced if the control dynamics can be designed to better condition the inverse computations.

The true test of dynamic inversion will be whether or not it can be successfully scaled up to larger tasks (Bartunov et al., 2018; Xiao et al., 2018). Even so, it may be useful in other contexts where it is necessary to invert a computation, such as motor control (Kawato, 1990) and sensory perception (Pizlo, 2001). As an example, it was pointed out in a recent paper (Vértes and Sahani, 2019) that the successor representation — used in reinforcement learning and requiring an inverse to calculate explicitly — can be achieved dynamically in a similar way to what we propose here.

## Broader Impact

The broader impacts of this work can be divided into two parts. First, there is the impact of the dynamic inversion algorithm on applications in machine learning and biological learning. Due to the costly nature of simulating such an inversion, we do not foresee its widespread use in training large-scale deep networks for applications. However, this study could have an impact on our understanding of learning in the brain at a very basic level, one day leading to implications for neurological disease, and also more neuroscience-related applications such as brain-machine interfaces. We foresee such results in the neuroscience and medical fields as primarily beneficial — any new insights into how the brain learns have the possibility to help those with disabilities or disorders.

Second, there is the broader impact of dynamic inversion as a general algorithm for inverting computations. Due to the widespread use of inverse computations in many domains and applications, it may be the case that the insights provided in this paper have an effect. For example, the development of better inverse models in robots could lead to substantial improvement for robotic applications, which would have a tremendous effect on society, likely both for better and worse. Overall, however, the work presented here is at the "basic research" level, and very far removed from specific applications.

## Acknowledgments and Disclosure of Funding

We thank members of the Machens lab for helpful comments and feedback. We also thank Alexander Meulemans for an enlightening discussion about approximate inversions and the relation of our work to target propagation. This work was supported by the Simons Collaboration on the Global Brain

(543009) and the Fundação para a Ciência e a Tecnologia (FCT; 032077). The authors declare no competing financial interests.

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
