[Supplementary Material]

# Supplementary Materials: Biological Credit Assignment through Dynamic Inversion of Feedforward Networks

**William F. Podlaski**
Champalimaud Research
Champalimaud Centre for the Unknown
1400-038 Lisbon, Portugal

**Christian K. Machens**
Champalimaud Research
Champalimaud Centre for the Unknown
1400-038 Lisbon, Portugal

## 7 Supplementary materials

### 7.1 Accuracy of dynamic inversion depends on dimensionality and controller leak

Accurate convergence of the backward pass dynamics is crucial for the success of dynamic inversion. We observe from Eq. (11) that the steady state of $\boldsymbol{\delta}_{l-1}$ depends upon the control signal for $\alpha > 0$. Furthermore, the nonlinear case in Eq. (16) features an implicit inversion of the nonlinearity, which also may have nontrivial effects on the steady state. Supplementary figure S1 illustrates the accuracy of dynamic inversion for different nonlinearities, relative dimensionalities, and leak values. We observe that accuracy degrades gradually for increasing leak values. The dynamics are most accurate for contracting networks, and least accurate for expanding networks, which require $\alpha > 0$. Networks with ReLU nonlinearities appear to be more inaccurate, but we attribute this to stability issues.

Figure S1: Evaluation of dynamic inversion accuracy for small example networks of dimensionality $20 \times 20$ (equal dim., left), $20 \times 10$ (contracting, middle), and $10 \times 20$ (expanding, left) for linear, tanh and ReLU transfer functions. Values for $\boldsymbol{\delta}_l$ were generated from a standard normal distribution, and dynamic inversion was run to obtain values for $\boldsymbol{\delta}_{l-1}$. Accuracy was measured as the angle (in degrees) between vectors $\tilde{\boldsymbol{\delta}}_l$ and $\boldsymbol{\delta}_l$ (top), as well as $\boldsymbol{\delta}_{l-1}$ and the explicit inversion $g^{-1}(\mathbf{M}_l^{DI}\boldsymbol{\delta}_l)$ (bottom), with $\mathbf{M}_l^{DI}$ as in Eq. (17). Note that the accuracy of $\boldsymbol{\delta}_{l-1}$ is not measured directly for ReLU because it does not have an explicit inversion. Weight matrices $\mathbf{W}_l$ and $\mathbf{B}_l$ were generated from random uniform distributions, and then $\mathbf{B}_l$ was optimized with SSA (see section 7.5 below). A total of 10 matrix initializations each with 10 generated values of $\boldsymbol{\delta}_l$ were run for each case.

## 7.2 Alternative control architecture for expanding layers

As noted in Section 3.2, the eigenvalues of the matrix $(\mathbf{W}_l\mathbf{B}_l - \alpha\mathbb{I})$ provide a good measure of the linear stability of dynamic inversion (true linear stability is measured by the eigenvalues of the block matrix in Eq. (S24) below). This precludes stability for $\alpha = 0$ and $d_l > d_{l-1}$ (expanding layer), as the matrix product $\mathbf{W}_l\mathbf{B}_l$ will be singular. To address this, we propose an alternative control architecture:

$$\dot{\boldsymbol{\delta}}_{l-1}(t) = -\alpha\boldsymbol{\delta}_{l-1}(t) + \mathbf{B}_l\boldsymbol{\delta}_l - \mathbf{B}_l\tilde{\boldsymbol{\delta}}_l(t) = -\alpha\boldsymbol{\delta}_{l-1}(t) + \mathbf{B}_l\mathbf{e}_l(t) \tag{S1}$$

$$\dot{\tilde{\boldsymbol{\delta}}}_l(t) = -\tilde{\boldsymbol{\delta}}_l(t) + \mathbf{W}_l g(\boldsymbol{\delta}_{l-1}(t)), \tag{S2}$$

where now the target error for layer $l - 1$, $\boldsymbol{\delta}_{l-1}$, integrates the error between $\boldsymbol{\delta}_l$ and $\tilde{\boldsymbol{\delta}}_l$ directly, through the feedback matrix $\mathbf{B}_l$. This can be interpreted as proportional feedback control with a fast controller. In this system, stability instead depends on the matrix $\mathbf{B}_l\mathbf{W}_l - \alpha\mathbb{I}$ (assuming the readout dynamics are fast, and so $\tilde{\boldsymbol{\delta}}_l = \mathbf{W}_l\boldsymbol{\delta}_{l-1}$). Note that this scheme either requires identical feedback weights for the target error $\boldsymbol{\delta}_l$ and the current estimate $\tilde{\boldsymbol{\delta}}_l$, or a separate population which calculates the error between these, propagated back as $\mathbf{B}_l(\tilde{\boldsymbol{\delta}}_l - \boldsymbol{\delta}_l) = \mathbf{B}_l\mathbf{e}_l$. This second option would feature error-coding units in the backward pass, similar to predictive coding (Whittington and Bogacz, 2019).

Figure S2: Schematic of alternate architecture for backward pass of dynamic inversion.

## 7.3 Single-loop dynamic inversion

Here we provide more details for single-loop dynamic inversion (SLDI). We first describe the backward pass dynamics for some simple example networks, and then discuss the more general case. We consider a network with $L = 3$ layers. In the backward pass, our aim is to begin with the output error, $\boldsymbol{\delta}_3$, and to arrive at the error for the two hidden layers, $\boldsymbol{\delta}_1$ and $\boldsymbol{\delta}_2$, simultaneously. Single-loop inversion dictates that we have a single feedback loop from the output (controller $\mathbf{u}_3(t)$) to the first hidden layer, with weight matrix $\mathbf{B}$. Considering a nonlinear controller, the dynamics will then be

$$\dot{\boldsymbol{\delta}}_1(t) = -\boldsymbol{\delta}_1(t) + \mathbf{B}\mathbf{u}_3(t) \tag{S3}$$

$$\dot{\boldsymbol{\delta}}_2(t) = -\boldsymbol{\delta}_2(t) + \mathbf{W}_2 g(\boldsymbol{\delta}_1(t)) \tag{S4}$$

$$\dot{\mathbf{u}}_3(t) = -\alpha\mathbf{u}_3(t) + \mathbf{W}_3 g(\boldsymbol{\delta}_2(t)) - \boldsymbol{\delta}_3. \tag{S5}$$

If the network features equal or expanding layer-by-layer dimensionalities ($d_1 \geq d_2 \geq d_3$), then we can describe the steady states of the two errors $\boldsymbol{\delta}_2$ and $\boldsymbol{\delta}_1$ as

$$\boldsymbol{\delta}_2 = g^{-1}(\mathbf{W}_3^+(\boldsymbol{\delta}_3 + \alpha\mathbf{u}_3)) \tag{S6}$$

$$\boldsymbol{\delta}_1 = g^{-1}(\mathbf{W}_2^+\boldsymbol{\delta}_2) = g^{-1}(\mathbf{W}_2^+ g^{-1}(\mathbf{W}_3^+(\boldsymbol{\delta}_3 + \alpha\mathbf{u}_3))), \tag{S7}$$

where we have used the fact that each weight matrix has a well-defined left (pseudo)-inverse. When the controller has no leak ($\alpha = 0$), the steady states here are equivalent to performing dynamic inversion sequentially, layer by layer. This generalizes to more hidden layers ($L > 3$) in a straightforward way.

In the case that the network has contracting layer-by-layer dimensionality ($d_1 < d_2 < d_3$), but is linear, we can also describe the steady state. This corresponds to a linear version of the architecture used in the nonlinear regression experiment, for which we test SLDI. Here, we can write the steady states for $\boldsymbol{\delta}_2$ and $\boldsymbol{\delta}_1$ as

$$\boldsymbol{\delta}_2 = \mathbf{B}(\mathbf{W}_3\mathbf{W}_2\mathbf{B})^{-1}\boldsymbol{\delta}_3 \tag{S8}$$

$$\boldsymbol{\delta}_1 = \mathbf{B}(\mathbf{W}_2\mathbf{B})^{-1}\boldsymbol{\delta}_2 = \mathbf{B}(\mathbf{W}_2\mathbf{B})^{-1}\mathbf{B}(\mathbf{W}_3\mathbf{W}_2\mathbf{B})^{-1}\boldsymbol{\delta}_3. \tag{S9}$$

For comparison, we also write the corresponding sequential DI steady state, in which each pair of hidden layers have feedback weights $\mathbf{B}_2$ and $\mathbf{B}_3$. In this case, it is

$$\boldsymbol{\delta}_2 = \mathbf{B}_3(\mathbf{W}_3\mathbf{B}_3)^{-1}\boldsymbol{\delta}_3 \tag{S10}$$

$$\boldsymbol{\delta}_1 = \mathbf{B}_2(\mathbf{W}_2\mathbf{B}_2)^{-1}\boldsymbol{\delta}_2 = \mathbf{B}_2(\mathbf{W}_2\mathbf{B}_2)^{-1}\mathbf{B}_3(\mathbf{W}_3\mathbf{B}_3)^{-1}\boldsymbol{\delta}_3. \tag{S11}$$

We thus see that the two solutions are distinct in this case, though similarities may still persist depending upon how the feedback weights are initialized. For example, in the nonlinear regression experiment shown in the main text, we initialize the SLDI feedback as $\mathbf{B} = -\mathbf{B}_1\mathbf{B}_2$, where $\mathbf{B}_1$ and $\mathbf{B}_2$ are the feedback matrices for sequential DI. For the negative transpose initialization, this makes $\mathbf{B} = -\mathbf{W}_1^T\mathbf{W}_2^T$. Generalizing SLDI to arbitrary dimensionalities, nonlinearities, and nonzero controller leak makes the steady state errors and relationship to sequential dynamic inversion less clear. Empirical evidence in the nonlinear regression experiment in the main text suggests a rough correspondence can persist between SLDI and sequential DI even in the general case (Fig. 3k), but more work is needed to examine this more closely.

### 7.3.1   Linearized case

Considering the alternative treatment of nonlinearities, we can define the SLDI dynamics as

$$\dot{\boldsymbol{\delta}}_1(t) = -\boldsymbol{\delta}_1(t) + \mathbf{B}\mathbf{u}(t) \tag{S12}$$

$$\dot{\boldsymbol{\delta}}_2(t) = -\boldsymbol{\delta}_2(t) + \mathbf{W}_2(\boldsymbol{\delta}_1(t) \oslash g'(\mathbf{a}_1)) \tag{S13}$$

$$\dot{\mathbf{u}}(t) = \mathbf{W}_3(\boldsymbol{\delta}_2(t) \oslash g'(\mathbf{a}_2)) - \boldsymbol{\delta}_3, \tag{S14}$$

where $\oslash$ indicates element-wise division. This somewhat peculiar handling of nonlinearities is to obtain the same formalism as the layer-wise dynamic inversion described in the main text. For equal or expanding layers, the steady states of the two errors $\boldsymbol{\delta}_2$ and $\boldsymbol{\delta}_1$ will be

$$\boldsymbol{\delta}_2 = \mathbf{W}_3^+(\boldsymbol{\delta}_3 + \alpha\mathbf{u}_3) \circ g'(\mathbf{a}_2) \tag{S15}$$

$$\boldsymbol{\delta}_1 = \mathbf{W}_2^+\boldsymbol{\delta}_2 \circ g'(\mathbf{a}_1) = \mathbf{W}_2^+(\mathbf{W}_3^+(\boldsymbol{\delta}_3 + \alpha\mathbf{u}_3) \circ g'(\mathbf{a}_2)) \circ g'(\mathbf{a}_1). \tag{S16}$$

Once again, when the controller has no leak, this will produce the same steady state as sequential dynamic inversion.

### 7.4   Relation of dynamic inversion to Gauss-Newton optimization

As mentioned in the main text, we suspect that dynamic inversion may relate to second-order methods. We study a simple case here as an illustration, and leave a more thorough analysis for future work. Following the derivations in Botev et al. (2017), we can write the block-diagonal sample Gauss-Newton (GN) matrix for a particular layer $l$ as

$$\mathbf{G}_l = \mathcal{Q}_l \otimes \mathcal{G}_l, \tag{S17}$$

where $\mathcal{Q}_l = \mathbf{h}_{l-1}\mathbf{h}_{l-1}^T$ is the sample input covariance to layer $l$ and $\mathcal{G}_l$ is the "pre-activation" GN matrix, defined recursively as

$$\mathcal{G}_l = \mathbf{D}_l\mathbf{W}_{l+1}^T\mathcal{G}_{l+1}\mathbf{W}_{l+1}\mathbf{D}_l = \mathbf{D}_l\mathbf{W}_{l+1}^T\mathbf{C}_{l+1}\mathbf{C}_{l+1}^T\mathbf{W}_{l+1}\mathbf{D}_l, \tag{S18}$$

with $\mathbf{D}_l = \text{diag}(g'(\mathbf{a}_l))$, and $\mathbf{C}_l$ is a square-root representation of $\mathcal{G}_l$. The GN update to the weight matrix of layer $l$ can be written in vectorized form as $\Delta\text{vec}(\mathbf{W}_l) \propto -\mathbf{G}_l^{-1}\mathbf{g}$, where $\mathbf{g}$ is a vectorized version of the standard backprop gradient, as in Eq. (2). In order to avoid vectorization (and thus simplify the comparison with dynamic inversion), we make the assumption that the input to this layer is whitened, making $\mathcal{Q}_l = \mathbb{I}_l$. This allows us to write the GN update in non-vectorized form:

$$\Delta\mathbf{W}_l \propto -\mathcal{G}_l^{-1}\boldsymbol{\delta}_l\mathbf{h}_{l-1}^T = (\mathbf{D}_l\mathbf{W}_{l+1}^T\mathbf{C}_{l+1}\mathbf{C}_{l+1}^T\mathbf{W}_{l+1}\mathbf{D}_l)^{-1}\mathbf{D}_l\mathbf{W}_{l+1}\boldsymbol{\delta}_{l+1}\mathbf{h}_{l-1}^T \tag{S19}$$

$$\approx (\mathbf{C}_{l+1}^T\mathbf{W}_{l+1}\mathbf{D}_l)^+\mathbf{C}_{l+1}^+\boldsymbol{\delta}_{l+1}\mathbf{h}_{l-1}^T. \tag{S20}$$

Note that the approximate equality is due to the assumption that both $(\mathbf{C}_{l+1}^T\mathbf{W}_{l+1}\mathbf{D}_l)$ has a left pseudoinverse, and $\mathbf{C}_{l+1}$ has a right pseudoinverse, which depends on the relative dimensionality of layers $l$ and $l+1$. Considering the simplest case, optimizing the penultimate set of weights $\mathbf{W}_{L-1}$

for a network solving regression with squared error loss, we have $\mathcal{G}_L = \mathbb{I}$ (and thus $\mathbf{C}_L = \mathbb{I}$), and $\boldsymbol{\delta}_L = \mathbf{e}$, and so the update becomes

$$\Delta\mathbf{W}_{L-1} \propto -(\mathbf{W}_L\mathbf{D}_{L-1})^+\mathbf{e}\mathbf{h}_{L-1}^T. \tag{S21}$$

The equivalent update for dynamic inversion with a nonlinear controller would be

$$\Delta\mathbf{W}_{L-1} \propto -g^{-1}(\mathbf{W}_L^+\mathbf{e})\mathbf{h}_{L-1}^T, \tag{S22}$$

where we see a clear resemblance. The main difference arises from how the nonlinearity appears — in dynamic inversion it is handled implicitly in the dynamics, whereas for layer-wise Gauss-Newton optimization, it is linearized. If we instead consider the linearized version of dynamic inversion, we would obtain

$$\Delta\mathbf{W}_{L-1} \propto -\mathbf{D}_{L-1}\mathbf{W}_L^+\mathbf{e}\mathbf{h}_{L-1}^T, \tag{S23}$$

where we now have a similar term to Eq. (S21), but it is not inverted. This suggests that a closer relationship between DI and GN optimization could be obtained by performing element-wise division by the gradient of the nonlinearity instead of multiplication (this would also transform the element-wise division in Eqs. (S13) and (S14) into multiplication). In practice, we found such treatment of nonlinearities to be unstable, and did not explore them in depth here.

A more thorough analysis is merited on the relationship between Eqs. (S21), (S22), and (S23) (as well as the types of nonlinear inversions found in (S22) and a more general comparison of dynamic inversion and GN optimization). We note that layer-wise whitening is performed in a recent model proposing to map natural gradient learning onto feedforward networks (Desjardins et al., 2015), suggesting that the strategic placement of whitening transformations in a network with dynamic inversion may produce a more accurate approximation to Gauss-Newton or natural gradient optimization. As mentioned in the main text, a more promising avenue would likely be to map dynamic inversion onto target propagation, where recent work has shown a more clear relationship to second-order learning (Meulemans et al., 2020; Bengio, 2020).

## 7.5 Stability optimization

In general, the dynamic inversion system dynamics for a particular layer $l$ are not stable when initialized with random matrices $\mathbf{W}_l$ and $\mathbf{B}_l$ (R-Init). We follow procedures outlined in Vanbiervliet et al. (2009) and Hennequin et al. (2014) to optimize linear stability by minimizing the smoothed spectral abscissa (SSA; a smooth relaxation of the spectral abscissa, the maximum real eigenvalue). The full system matrix can be written in block form as

$$\mathbf{M}_l = \begin{bmatrix} -\mathbb{I} & \mathbf{B}_l \\ \mathbf{W}_l & -\alpha\mathbb{I} \end{bmatrix}, \tag{S24}$$

with the first and second rows corresponding to the dynamics of $\boldsymbol{\delta}_{l-1}$ and $\mathbf{u}_l$, respectively, of Eqs. (5) and (10). In brief, we calculate the gradient of the SSA with respect to the matrix $\mathbf{B}_l$, and make small steps until the maximum eigenvalue is sufficiently negative. We refer the reader to the references above for details. SSA optimization can be done both on $\mathbf{W}_l$ and $\mathbf{B}_l$, but we chose to optimize only $\mathbf{B}_l$ in order to have full control on the initialization of $\mathbf{W}_l$.

## 7.6 Algorithms for dynamic inversion

We provide pseudocode for recursively calculating the backpropagated error signals ($\boldsymbol{\delta}_l$) for dynamic inversion (DI) and non-dynamic inversion (NDI), including the different schemes introduced in Fig. 2. Following the calculation of the error signals, weights and biases are updated according to the standard backpropagation rules (Eq. (2)).

---

**Algorithm 1** Dynamic Inversion (Sequential, nonlinear controller)

---

**function** DYN-INV $(\mathbf{W}_l, \mathbf{B}_l, g(\cdot), \alpha, \boldsymbol{\delta}_l, \mathrm{dt}, \mathrm{tsteps})$:
    $\boldsymbol{\delta}_{l-1}, \mathbf{u}_l \leftarrow \mathbf{0}$
    **for** $t = 1$ to tsteps **do**
        $\boldsymbol{\delta}_{l-1}\mathrel{+}= \mathrm{dt}(-\boldsymbol{\delta}_{l-1} + \mathbf{B}_l\mathbf{u}_l)$
        $\mathbf{u}_l \mathrel{+}= \mathrm{dt}(-\alpha\mathbf{u}_l + \mathbf{W}_l g(\boldsymbol{\delta}_{l-1}) - \boldsymbol{\delta}_l)$
    **end for**
    **return** $\boldsymbol{\delta}_{l-1}$

---

---
**Algorithm 2** Dynamic Inversion (Sequential, linearized controller)
---

   **function** DYN-INV $(\mathbf{W}_l, \mathbf{B}_l, g'(\mathbf{a}_{l-1}), \alpha, \boldsymbol{\delta}_l, \text{dt}, \text{tsteps})$:
      $\boldsymbol{\delta}_{l-1}, \mathbf{u}_l \leftarrow \mathbf{0}$
      **for** $t = 1$ to tsteps **do**
         $\boldsymbol{\delta}_{l-1} \mathrel{+}= \text{dt}(-\boldsymbol{\delta}_{l-1} + \mathbf{B}_l \mathbf{u}_l)$
         $\mathbf{u}_l \mathrel{+}= \text{dt}(-\alpha \mathbf{u}_l + \mathbf{W}_l \boldsymbol{\delta}_{l-1} - \boldsymbol{\delta}_l)$
      **end for**
      **return** $\boldsymbol{\delta}_{l-1} \circ g'(\mathbf{a}_{l-1})$

---

---
**Algorithm 3** Two-Layer Dynamic Inversion (Repeat hidden layers, nonlinear controller)
---

   **function** REP-2L-DYN-INV $(\mathbf{W}_{l-1}, \mathbf{W}_l, \mathbf{B}_{l-1}, \mathbf{B}_l, g_l(\cdot), g_{l-1}(\cdot), \alpha_l, \alpha_{l-1}, \boldsymbol{\delta}_l, \text{dt}, \text{tsteps})$:
      $\boldsymbol{\delta}_{l-1}, \boldsymbol{\delta}_{l-2}, \mathbf{u}_l, \mathbf{u}_{l-1} \leftarrow \mathbf{0}$
      **for** $t = 1$ to tsteps **do**
         $\boldsymbol{\delta}_{l-1} \mathrel{+}= \text{dt}(-\boldsymbol{\delta}_{l-1} + \mathbf{B}_l \mathbf{u}_l)$
         $\mathbf{u}_l \mathrel{+}= \text{dt}(-\alpha_l \mathbf{u}_l + \mathbf{W}_l g_l(\boldsymbol{\delta}_{l-1}) - \boldsymbol{\delta}_l)$
         $\boldsymbol{\delta}_{l-2} \mathrel{+}= \text{dt}(-\boldsymbol{\delta}_{l-2} + \mathbf{B}_{l-1} \mathbf{u}_{l-1})$
         $\mathbf{u}_{l-1} \mathrel{+}= \text{dt}(-\alpha_{l-1} \mathbf{u}_{l-1} + \mathbf{W}_{l-1} g_{l-1}(\boldsymbol{\delta}_{l-2}) - \boldsymbol{\delta}_{l-1})$
      **end for**
      **return** $(\boldsymbol{\delta}_{l-1}, \boldsymbol{\delta}_{l-2})$

---

---
**Algorithm 4** Two-Layer Dynamic Inversion (Single loop, nonlinear controller)
---

   **function** SL-2L-DYN-INV $(\mathbf{W}_{l-1}, \mathbf{W}_l, \mathbf{B}, g_l(\cdot), g_{l-1}(\cdot), \alpha, \boldsymbol{\delta}_l, \text{dt}, \text{tsteps})$:
      $\boldsymbol{\delta}_{l-1}, \boldsymbol{\delta}_{l-2}, \mathbf{u}_l \leftarrow \mathbf{0}$
      **for** $t = 1$ to tsteps **do**
         $\boldsymbol{\delta}_{l-2} \mathrel{+}= \text{dt}(-\boldsymbol{\delta}_{l-2} + \mathbf{B} \mathbf{u}_l)$
         $\boldsymbol{\delta}_{l-1} \mathrel{+}= \text{dt}(-\boldsymbol{\delta}_{l-1} + \mathbf{W}_{l-1} g_{l-1}(\boldsymbol{\delta}_{l-2}))$
         $\mathbf{u}_l \mathrel{+}= \text{dt}(-\alpha_l \mathbf{u}_l + \mathbf{W}_l g_l(\boldsymbol{\delta}_{l-1}) - \boldsymbol{\delta}_l)$
      **end for**
      **return** $(\boldsymbol{\delta}_{l-1}, \boldsymbol{\delta}_{l-2})$

---

---
**Algorithm 5** Error propagation for BP, FA, PBP, and NDI
---

   **function** ERR-INV $(\text{algo}, \mathbf{W}_l, \mathbf{B}_l, g'(\mathbf{a}_{l-1}), \alpha, \boldsymbol{\delta}_l)$:
      $\mathbf{M}_l = \textbf{switch}(\text{algo})$:
         **case**(BP): $\mathbf{W}_l^T$
         **case**(FA): $\mathbf{B}_l$
         **case**(PBP): $\mathbf{W}_l^+$
         **case**(NDI):
            **if** $d_l > d_{l-1}$ **then**
               $(\mathbf{B}_l \mathbf{W}_l - \alpha \mathbb{I})^{-1} \mathbf{B}_l$
            **else**
               $\mathbf{B}_l (\mathbf{W}_l \mathbf{B}_l - \alpha \mathbb{I})^{-1}$
            **end if**

      $\boldsymbol{\delta}_{l-1} \leftarrow \mathbf{M}_l \boldsymbol{\delta}_l \circ g'(\mathbf{a}_{l-1})$
      **return** $\boldsymbol{\delta}_{l-1}$

---

| Experiment | Architecture | Nonlinearities | Training Iters | Learning Rate | Weight Decay | Leak ($\alpha$) |
|---|---|---|---|---|---|---|
| Linear regression | 30-20-10 | linear-linear | 2000 | $10^{-2}$ | 0 | 0 (blue); $10^{-3}$ (yellow); $10^{-2}$ (red) |
| Nonlinear regression | 30-20-10-10 | tanh-tanh-linear | 12000 | $10^{-2}$ | 0 | 0 (blue); $10^{-3}$ (yellow); $10^{-2}$ (red) |
| MNIST classification | 784-1000-10 | tanh-softmax | $6 \times 10^5$ (10 epochs) | $10^{-3}$ | $10^{-6}$ | 0 (blue); $10^{-2}$ (yellow) |
| MNIST autoencoding | 784-500-250-30 -250-500-784 | tanh-tanh-linear-tanh-tanh-linear | $1.6 \times 10^6$ (25 epochs) | $10^{-6}$ | $10^{-10}$ | $10^{-3}$ (blue); $10^{-2}$ (yellow) |

Table 1: Hyperparameters and architectures for all experiments.

## 7.7 Simulation details

### 7.7.1 General comments

Due to instabilities in the PBP and NDI algorithms during the first several iterations of training, we imposed a maximum norm for the backpropagated error signals for linear and nonlinear regression ($\|\boldsymbol{\delta}\| \leq 10$ for linear regression, $\|\boldsymbol{\delta}\| \leq 1$ for nonlinear regression). This was not necessary for MNIST classification or MNIST autoencoding. This did not affect BP and FA algorithms, and if anything places a handicap on the dynamic inversion algorithms. Stability was measured initially and throughout training by computing the maximum real eigenvalue for the block matrix in Eq. (S24).

### 7.7.2 Linear regression (Fig. 3a-f)

The linear regression example utilized a network with a single hidden layer (dim. 30-20-10) following Lillicrap et al. (2016). Training data was generated in the following way: input data $\mathbf{x}$ was generated independently for each dimension from a standard normal distribution, and target output $\mathbf{t}$ was generated by passing this input through a matrix $\mathbf{T}$, with elements generated randomly from a uniform distribution ($\mathcal{U}(-1, 1)$) such that $\mathbf{t} = \mathbf{T}\mathbf{x}$. No bias units were used in the network (nor to generate the test data). Weight matrices ($\mathbf{W}_0, \mathbf{W}_1$) were initialized with random uniform distributions ($\mathcal{U}(-0.01, 0.01)$) and all algorithms began with exact copies. The random feedback matrix ($\mathbf{B}$, R-Init) was generated from the same distribution, but for feedback alignment, this distribution had a larger spread ($\mathcal{U}(-0.5, 0.5)$ as in Lillicrap et al. (2016); negative transpose initialization for FA was also scaled: $\mathbf{B} = -50\mathbf{W}^T$). Training used squared error loss, and we plot the training error as normalized mean squared error (NMSE) in which the error for each algorithm is normalized by the maximum error across all algorithms and iterations, so that training begins with a normalized error of ~1. The learning rate was set to $10^{-2}$ for all algorithms and was not optimized.

### 7.7.3 Nonlinear regression (Fig. 3g-k)

The nonlinear regression example was also adapted from Lillicrap et al. (2016) and used a network with two hidden layers (dim. 30-20-10-10) and tanh nonlinearities (with linear output). Training data was generated from a network with the same architecture, but with randomly generated weights and biases (all generated from a uniform distribution, $\mathcal{U}(-0.01, 0.01)$). Feedforward and feedback weight matrices were initialized in the same way as the linear regression example, and bias weights were initialized to zero. Training loss was again squared error, and normalized in the same was as for linear regression. The learning rate was set to $10^{-3}$ for all algorithms and was not optimized.

### 7.7.4 MNIST classification (Fig. 4a)

MNIST classification was performed on a single hidden layer network (dim. 784-1000-10 as in Lillicrap et al. (2016)) with a tanh nonlinearity and a softmax output with cross-entropy loss. The

standard training (60000 examples) and test (10000 examples) sets were used. Data was first preprocessed by subtracting the mean from each pixel dimension and normalizing the variance (across all pixels) to 1. Weight matrices and biases were initialized in the same way as for linear and nonlinear regression. Training was performed online (no batches). The learning rate was set to $10^{-3}$ for all algorithms and was not optimized. An additional weight decay of $10^{-6}$ was also used.

### 7.7.5  MNIST autoencoder (Fig. 4b-d)

MNIST autoencoding was done on a network with architecture 784-500-250-30-250-500-784 with nonlinearities tanh-tanh-linear-tanh-tanh-linear, similar to Hinton and Salakhutdinov (2006) but with one hidden layer removed. The standard MNIST training set was used (60000 examples), and performance was measured on this dataset, without the use of the test set. Data was preprocessed so that each pixel dimension was between 0 and 1 (data was not centered). To speed up simulations, training was done on mini-batches of size 100. The learning rate was set to $10^{-6}$ for all algorithms, with a weight decay of $10^{-10}$. Learning rate and mini-batch size were not optimized, however, we found that PBP and NDI algorithms became unstable for larger learning rates.

### 7.8  Code

Code for running dynamic inversion, and for reproducing the experiments shown here can be found at `https://github.com/wpodlaski/dynamic-inversion`.