[Reviews · NeurIPS 2020]

Review 1

Summary and Contributions: The authors present a novel way using feedback control to calculate backpropagated errors using feedback control. They claim that this new algorithm is more biologically plausible because it solves the weight transport problem.

Strengths: See below.

Weaknesses: See below.

Correctness: Yes.

Clarity: Very clearly written.

Relation to Prior Work: Yes.

Reproducibility: Yes

Additional Feedback: Update: I appreciate the authors' response. I raise my score trusting that they will discuss the points they raised in the rebuttal also in the final paper. ------------ ------------ I enjoyed reading this paper, using feedback control to perform local credit assignment is an interesting and a novel idea. I do have some concerns: 1) Of the many biologically implausible features of backpropagation, this paper solves only the weight transport problem, for which, as the authors have pointed, many other solutions have been proposed. The advantage of this method is explained to be "Unlike many other biologically-plausible algorithms for credit assignment, dynamic inversion does not require precise feedback weights. This is a crucial distinction, as it not only relaxes the assumptions on feedback wiring, but also allows for feedback to be used concurrently for other roles, such as attention and prediction." However, how feedback could be switching between these different modes of operation is not explained. Therefore, this claim is unjustified. A related point is how biological circuits can shut down the feedback circuit during the forward pass. 2) Solving the weight transport problem is not sufficient to call an algorithm "biological". To implement feedback control, authors introduce biologically questionable architectures given in Figure 2. Therefore, it is not clear to me that this new scheme is more biologically-plausible than the original backpropagation. I am willing to change my score if the authors can provide a biological justification of how architectures in Figure 2 can be implemented. On another note, there is another straight-forward and biologically-plausible solution to the weight transport by separate Hebbian learning of feedforward and feedback weights as in Kolen, John F., and Jordan B. Pollack. "Backpropagation without weight transport." Authors should discuss this simple solution.


Review 2

Summary and Contributions: This paper proposes a mechanism by which error signals can be propagated backward through a neural network in order to perform useful weight updates. The authors show that feedback control dynamics can ensure, without requiring exact weight symmetry between forward weights and backward error weights, that the error signal is approximately transformed by the pseudoinverse of the forward weights at each layer. They also provide some experiments testing an implementation of this approach.

Strengths: The authors make an interesting observation, that dynamics can enforce a particular feedback transformation without precisely tuning the feedback weights to give that transformation directly. Their mathematical claims are correct, as far as I can tell. The biological implications of this idea are interesting.

Weaknesses: Major points 1. As the authors note, the stability of the feedback dynamics depends on a condition on the eigenvalues of WB - alpha*I. Without it, the feedback dynamics will yield unpredictable results and presumably not perform effective credit assignment. This condition is extremely unlikely to be satisfied generically, and is essentially the analog of sign-symmetry in forward and backward weights when one considers pseudoinverses rather than transposes. The authors manually enforce that it be satisfied at initialization, and manually adjust the backward weights if the condition is violated during training. These manual initialization choices and adjustments are doing much of the work of credit assignment in the authors' algorithm -- I can't tell from the results as presented how helpful the dynamic inversion really is. For instance, how well would it work if you used the same weight initialization and stability correction procedure but did not run any dynamics? 2. Even ignoring issue #1, the empirical results are a mixed bag. The results on the regression tasks seem good, but the fact that (the non-dynamic approximation to) the authors' algorithm performs worse than feedback alignment on MNIST classification is concerning. The initialization dependence observed in the autoencoding experiments is interesting, but probably needs further study -- it appears that in this problem, both backprop and NDI are fairly sensitive to the initialization, but with opposite preferences. Given that backprop is the workhorse of much more complex networks than these, it would require substantial evidence to show convincingly that inversion has a consistent advantage over backprop in initialization to robustness. 3. It would be good to see the actual proposed method implemented in the MNIST experiments rather than the non-dynamic "ideal" version. One of the obvious concerns with a method like this is the convergence of the dynamics, and right now it is not possible to tell how the speed of convergence will scale to harder problems. Of course, to really answer this question would require going beyond MNIST, but showing the dynamic results on MNIST would be a start.

Correctness: The discussion of the algorithmic details and the math behind them appears correct. I don't see any obvious problems with the experiments that were run, but see above for some ablations / experiments I think would be important to see in order to justify the paper's claims.

Clarity: Yes, it is presented clearly. I would have benefitted from a bit more discussion of the consequences of using SDI vs. DI. I also think the presentation does not appropriately emphasize the importance of stability to the algorithm and the non-obviousness of how it might be achieved in biologically plausible fashion.

Relation to Prior Work: Yes, I think so. The paper makes the appropriate connections with the existing bio-plausible credit assignment literature, and with the feedback control literature that inspires the method.

Reproducibility: Yes

Additional Feedback: For me the stability issue is very significant, as the proposed method introduces a new problem (maintaining sign of eigenvalues of WB - alpha*I) that is roughly as difficult as the standard weight transport problem (maintaining approximate symmetry between W and B). I think it is important either to propose and test a bio-plausible mechanism of enforcing stability, or else to show empirically that the dynamic inversion mechanism provides substantial benefit beyond what you get for free by enforcing the stability condition. ******************** Update after author response: Thanks to the authors for clarifying that the stability condition typically remained satisfied without explicit enforcement. This suggests to me that the symmetric initialization is doing a lot of work for the method. I think it would be valuable / clarifying for readers of this paper to see a comparison between the paper's method and a baseline that uses feedback alignment with the same symmetric initialization. This comparison would make clear how much the core component of the proposed method (inversion through dynamics) is contributing to the results vs. how much is contributed by other auxiliary implementation details like the initialization. All that said, the authors note in the response that the most important contribution of this paper is to introduce the idea of inversion through dynamics in the context of credit assignment, and so the empirical results are of secondary importance. I agree, and I think the idea is cool enough and new (to my knowledge) enough that I'd vote for acceptance.


Review 3

Summary and Contributions: The authors propose a scheme for biologically plausible credit assignment in multilayer networks based on inverting the forward computation of each layer through an iterative control mechanism (which settles to some form of inverse of the deltas of the next layer to form the deltas of the previous layer). Note that these deltas are not the usual error gradients, and the authors attempt to connect them with Gauss-Newton updates, but the link remains to be clarified. The paper also shows the results of several small scale experiments similar to those done in the early paper on feedback alignment, showing that although the proposed scheme is very slow (and thus difficult to scale to something like ImageNet) it seems to work well in these smaller-scale settings.

Strengths: The paper explores a fairly original avenue for credit assignment (based on inversion of 'errors' rather than propagation of gradients). This is clearly relevant to NeurIPS theme of bridging the gap between biological neural networks and artificial ones. The empirical evaluation involves several datasets and architectures, although all at a small scale and not using state-of-the-art architectures for these datasets. Yet, this is expected for novel explorations in this difficult field of investigation, as a first stop to evaluate new approaches.

Weaknesses: I believe that this approach is really very interesting and promising but somewhat lacking a theoretical justification in terms of credit assignment. We know that gradient descent is a good thing to do in terms of optimization (and SGD has generalization properties as well). Although the proposed idea is intuitively appealing, it is much less clear what the proposed credit assignment scheme guarantees in terms of optimizing the output error. Expanding the paper in that direction would be useful, as previous explorations of biologically motivated learning rules have often worked well in small-scale settings but failed in more complex ones used in modern deep learning application (which are themselves tiny compared to what the brain is achieving). One intuition I have in this respect is that finding a delta for intermediate layer l such that applying the downstream feedforward computation to it would give the output error makes sense in a non-parametric setting (as if the layer was able to perform any computation). However, the layer is very limited in its capacity and parametrization (basically it is just a composition of affine and rectification) and so we only care about a target change which is 'feasible'. That is why backprop works: it proposes only a small change (in the direction of the gradient). This also motivates approaches like Equilibrium Propagation which look for small perturbations of each layer (to move towards a lower output error). It may be interesting to extend the paper's analysis to the limit case where we scale the output deltas by a small positive beta, so that we operate in this small variation regime which gets such algorithms maybe closer to gradient estimation. I believe this may work well.

Correctness: In general the methodology is correct. I am concerned with the 2% error rate of the MLP on MNIST since I believe that much better results (around 1%) can be obtained by simple changes in the architecture (such as using rectifiers and batchnorm). Even better results can be obtained with a deep MLP on MNIST, and this should be amenable to the kind of simulations required for the proposed methods. The claims about the links to Gauss-Newton are maybe too strong and need to be justified better or made clearer and more solid or the claims need to be made weaker (and simply point to a possibly interesting connection). The statement in line 101 may be incorrect: indeed if alpha is small (please show values used) it would reduce the inversion error, however, this is only true if u does not compensate for alpha being small by being corresponding large. And in fact if you look at eqn 8 and set it to 0 and isolate u, you see that it is going to be exactly inversely proportional to alpha: in other words, making alpha small has possibly no effect on alpha*u because u compensates linearly.

Clarity: Yes in general, well above average. The e in eqn 17 is not defined. The single-loop dynamic inversion (SDI) is not clearly described and it is the one used at the end of the day for several experiments, so this really needs to be taken care of.

Relation to Prior Work: Yes, good coverage of the prior literature and clear explanations of the differences.

Reproducibility: Yes

Additional Feedback: Please provide the values of alpha you end up using, and verify how good is the actual inversion (see my comment above about u's magnitude compensating for smaller alpha). To conclude, the biggest weakness with this paper is that it is not yet clear whether the proposed inversion-based credit assignment optimizes the parameters in a formal sense. The only strong (positive) clue comes from the experiments (and a handwaving argument which could be strengthened). Just compare eqn 16 with the backprop equations. An important missing element is that a_l is not used at all in the process, which intuitively seems wrong to me. Thanks for the rebuttal, I maintain my acceptance recommendation.


Review 4

Summary and Contributions: This paper proposes a novel method for implementing a target-prop backwards pass in MLPs which does not require learning inverse weights but instead dynamically computes the inverse through the use of a PID controller (they used integral only control). They test their method on a linear and nonlinear regression task and on MNIST. This is a good paper and was righty accepted.

Strengths: Their dynamic inversion method is novel and interesting. In general the idea of dynamically solving for inverses instead of learning them is interesting and could lead to new insights. The mathematical and theoretical grounding is strong. If the algorithm proves scalable the contribution may be significant.

Weaknesses: As mentioned in the paper, the primary limitation is the scalability of the method, especially to more complex tasks than mnist. The cost of the dynamical inversion step relative to target-prop is also worrisome. The method is novel though and I think it is worthy of publication. The primary weakness is only experimental evaluation on MNIST. This is sufficient I believe for a first paper but further work must investigate the efficacy of this method on more difficult tasks. Also, they should evaluate the full dynamic inversion on MNIST at least, regardless of computational cost. minor notes: Some of the connections between this target-prop like inverse methods and gauss newton have been previously covered in prior work. The axis scale on figure 4a (mnist test accuracy) is terrible and only really tells you that accuracy is above 80% which is not impressive for mnist.

Correctness: I believe the claims and empirical methodology are correct.

Clarity: The paper is well written and clear.

Relation to Prior Work: Most prior work is well and clearly discussed. One omission which is heavily relevant to the link between targetprop and gauss-newton optimisation is: https://scriptiebank.be/sites/default/files/thesis/2019-10/Master_Thesis_Alexander_Meulemans.pdf https://arxiv.org/pdf/2006.14331.pdf

Reproducibility: Yes

Additional Feedback: While I believe the method itself is likely too computationally expensive and unwieldy to be really scalable, I think it is a novel and interesting contribution to the literature, and the dynamical inversion idea may be a productive one. I think it should be published.

[Author Response · NeurIPS 2020]

We thank the reviewers for their constructive and fair criticism of our submission. Overall we noted four main areas of critique, addressed below with reference to each reviewer's specific comments (reviewers referred to as R1-4).

**1. Biological plausibility (R1):** *R1 questioned the biological plausibility of Fig 2, the mechanisms of switching or gating of feedback in the brain, and the concurrent use of feedback weights for other purposes.*

We insist that our claims of plausibility are reasonable. First, solving the weight transport problem alone should make any algorithm more plausible than backprop. Second, we hypothesize that the remaining issues with Fig 2 (signed errors, separate passes, connectivity gating) could be mitigated in combination with insights from other studies (e.g., propagation of targets, signal multiplexing with dendritic compartments), but are not central to dynamic inversion as a novel solution to weight transport. Furthermore, our hypotheses of feedback gating are not far from neuroscientific theories about inhibition and neuromodulation — e.g., acetylcholine may control the relative strengths of feedforward and feedback connectivity (Hasselmo 2006, "The role of acetylcholine in learning and memory."). We are willing to soften our claims of plausibility in the text, though we note that the term "biologically-plausible" is often loosely defined, and it is common in the literature to tackle one problem at a time. Lastly, in our statements of using feedback for other purposes, we simply wished to point out that this may lead to conflicts in weight values, and the flexibility of dynamic inversion could help (we can soften this claim too). We did not mean to claim that dynamic inversion is always better than learning feedback (e.g., Kolen & Pollack 1994), which we can elaborate on in the discussion.

**2. Stability and leak (R2,R3):** *R2 argued that stability is crucial and under-stressed; R3 suggested that the leak parameter $\alpha$ may have a large effect on the steady state.*

We find the questions raised about stability and leak to be of less concern than the reviewers fear. Stability was rarely (if ever) a problem in our experiments — only for linear regression was it necessary to perform stability optimization during training (and only during the first $\sim$100 iterations of 2000); for nonlinear regression, MNIST classification, and MNIST autoencoding, the dynamics remained stable once the initialization was set (also note that we keep track of and optimize stability for non-dynamic inversion too). Furthermore, it is both common in models and well within the hypothesized capabilities of biology to

specify precise initializations for learning algorithms (e.g., see Zador 2019, "A critique of pure learning..."). Thus, we contend that R2's suggestion of testing a plausible stability-enforcing mechanism is not critical here, and testing stability optimization without dynamics would be equivalent to feedback alignment with a particular weight initialization. Overall, we argue that stability enforcement (through dynamics and/or plasticity) seems reasonable given current neuroscientific theories (e.g., Zenke, Ganguli & Gerstner 2017, "The temporal paradox of Hebbian learning and homeostatic plasticity."). As for the leak and scaling of $\alpha\mathbf{u}_l$ — we can include simulations showing that the leak produces gradual increases in error (example on the right shows angle between $\boldsymbol{\delta}_l$ and output reconstruction $\tilde{\boldsymbol{\delta}}_l$).

**3. MNIST results and scalability (R2,R3,R4):** *R2-4 were concerned with performance on MNIST classification; R2 and R4 pointed out that only non-dynamic inversion (NDI) was implemented on the MNIST tasks; R2 and R3 wanted more explanation of single-loop inversion (SDI); R4 noted concerns of scalability in general.*

We stress that our aim in this submission was to introduce a proof-of-concept idea, and so we find the deficiencies in performance to be concerning, but not critical. As the reviewers suggest, fine-tuning the architecture, hyperparameters, and optimization procedure may change these results (as well as further study into the conditioning of the inversion). Furthermore, the results on MNIST autoencoding show potential benefits over feedback alignment. We justify the fact that only non-dynamic inversion (NDI) was used in the MNIST examples with the observation that NDI behaves nearly identically to DI (Fig 3d,i; when $\alpha > 0$), though we would agree to verify this. We also agree with the reviewers that not enough detail was given for single-loop inversion (SDI), and we would be happy to elaborate on it.

**4. Theoretical justification and link to Gauss-Newton (GN) optimization (R3,R4):** *R3 argued that the paper lacks theoretical justification, and questioned the link to GN optimization; R4 pointed out some omitted literature.*

We again reiterate that the main aim of the submission was to introduce a novel idea. While it is true that a theoretical understanding of our algorithm is lacking (and we thank R3 for the interesting suggestions), it is common for such rigorous theoretical work to follow publication of the original idea. For example, target propagation has recently received more rigorous analysis which may be applicable to dynamic inversion of targets, which we can cite (R4 mentioned Meulemans et al. 2020 "A theoretical framework for target propagation", and see Bengio 2020, "Deriving differential target propagation from iterating approximate inverses."). In any case, we do agree to soften our claims that dynamic inversion approximates GN optimization, and instead point it out as an interesting link for future study. Lastly, R3's mention that the pre-activation variable $\mathbf{a}_l$ doesn't appear in the update can be readily explained. In backpropagation, this variable is only needed to estimate the slope of the nonlinearity, $g'(\mathbf{a}_l)$. In dynamic inversion, the nonlinearity is included implicitly in the dynamics (which is arguably more biologically-plausible), though this may be problematic for approximate inversions and thus merits further study.

[Meta-Review · NeurIPS 2020]

The reviewers all agreed that this is a worthwhile contribution to the biologically realistic credit assignment problem and a consensus on "Accept" was achieved easily.